# Spatiotemporal Assessment of Induced Seismicity in Oklahoma: Foreseeable Fewer Earthquakes for Sustainable Oil and Gas Extraction?

**Zhen Hong [1], Hernan A. Moreno [1,2,*]** and **Yang Hong [2]**

1   Department of Geography and Environmental Sustainability, University of Oklahoma, Norman, OK 73019, USA; zhen.hong-1@ou.edu
2   School of Civil Engineering and Environmental Science, University of Oklahoma, Norman, OK 73019, USA; yanghong@ou.edu
*   Correspondence: moreno@ou.edu; Tel.: +1-480-399-0571

**Abstract:** In this study we present a spatiotemporal analysis of the recent seismicity and industry-related wastewater injection activity in Oklahoma. A parsimonious predictive tool was developed to estimate the lagged effect of previous month's injection volumes on subsequent regional seismic activity. Results support the hypothesis that the recent boom in unconventional oil and gas production and either the mitigation policies or the drop in oil prices (or both) are potentially responsible for the upsurge and reduction in the state's seismic activity between 2006–2015 and 2016–2017, respectively. A cluster analysis reveals a synchronous migration pattern between earthquake occurrences and salt water injection with a predominant northwest direction during 2006 through 2017. A lagged cross-correlation analysis allows extracting a power law between expected number of quakes and weighted average monthly injection volumes with a coefficient of determination of $R^2 = 0.77$. Such a relation could be used to establish "sustainable water injection limits" aiming to minimize seismicity to values comparable with several historically representative averages. Results from these analyses coincide on previously found sustainable limits of 5 to 6 million $m^3$/month but expand to operations that could attain the same number through differential monthly planning. Findings could potentially be used for model intercomparison and regulation policies.

**Keywords:** Oklahoma seismicity; wastewater injection; injection-induced earthquakes; sustainable oil and gas extraction

## 1. Introduction

Prior to the year 2000, the United States had an average of 21 earthquakes each year with magnitude 3.0 (i.e., Mw 3.0) or greater; however, since the start of 2010, more than 300 earthquakes of equal or greater magnitude occurred in three years [1]. In the U.S. Great Plains region, the rate of increase has been mostly attributed to excessive volumes of wastewater injection due to the unprecedented activity of the oil and gas industry [1–8]. For example, the Mw 3.9 earthquake on 31 December 2011 in Youngstown, Ohio was concluded to be induced by the fluid injection at a deep-injection well close to pre-existing faults [4]. The Mw 4.7 earthquake on 27 February 2011 in central Arkansas occurred within 6 km of three wastewater disposal wells in use [3]. The 2008–2009 sequence of earthquakes with Mw smaller than 3.3 at the Dallas/Fort Worth Airport area were potentially induced by brine disposal associated with the production of natural gas [2]. The Mw 5.7 and 5.8 earthquakes in 2011 and 2016 in Oklahoma appear to be relevant to wastewater injection [9,10]. However, to conclusively determine the degree of association between wastewater injection and

earthquakes remains a challenging task by the research limitations in data availability and regionally appropriate seismic models.

Both earthquakes' regional number and magnitude have increased during the current decade and seismic events have become common within the state of Oklahoma, including recorded earthquakes with Mw 3.0 or greater. Keranen et al. [7] noted that the total number of earthquakes in Oklahoma between 2008 and 2013 (i.e., 6 years) was four times those occurred from 1976 to 2007 (i.e., 31 years). Additionally, between 1974 and 2008, Oklahoma only had an average of one earthquake with Mw $\geq$ 3.0 each year. Comparatively, during 2013 and 2014, the state had more than 100 Mw $\geq$ 3.0 quake events per year [11]. In 3 September 2016, an earthquake with a Mw 5.8 occurred near the northern Oklahoma town of Pawnee. It was the strongest earthquake on record to date. As an immediate response, the Oklahoma Corporation Commission (OCC) ordered the shutdown of 37 disposal wells to within an area of 1878 km$^2$ of the epicenter. OCC has also taken many other actions in response to recent earthquakes, including a disposal volume reduction plan [12].

A growing body of scientific research increasingly connects this upsurge in seismic activity in Oklahoma with the recent boom in oil and gas production, specifically with the wastewater injection volumes (IW) and their depth [7,9–11,13–16]. Norbeck and Rubinstein [17] calibrated a reservoir model to calculate the hydrologic conditions associated with the activity of 902 saltwater disposal wells into the Aurbuckle aquifer using multiple geology parameters within Oklahoma. Langenbruch and Zoback [18] calibrated statistical model that relates seismicity and injection by applying the Guttenberg–Richterl Law depending on two varying-in-time and space seismogenic parameters. Despite recent contributions of robust, but parametrically uncertain, physically based and hybrid (statistics and physics-based) models [17–20], more work is needed to assess the cumulative effects of underground injected water on triggering subsequent seismic activity and to identify bivariate regional migration patterns that allow establishing clear relations between water injection and earthquake magnitude and number. This study has gathered exhaustive datasets pertaining to underground injection control (UIC) wells from OCC and the earthquake catalogue data from Oklahoma Geological Survey (OGS) from 2006 to 2017. The main findings of this article could help setting sustainable limits for oil and gas extraction industry in order to minimize the expected number and magnitude of induced quakes, thus avoiding future human and property losses.

This manuscript first provides a description of the data sources and magnitude of completeness to then develop spatiotemporal relations of the wastewater injection and seismic activity in Oklahoma during 2006–2017. Subsequently, it provides an assessment of the regional collocation of wastewater injection activity and number/magnitude of earthquakes and their spatial association. Then, it explores the temporal correlations between wastewater injection volumes and number of earthquakes to develop a 2-parameter (i.e., parsimonious) predictive power law. Model results are evaluated in terms of observations and two recently published models. A discussion section describes the potential uses and limitations of the achieved results. Lastly, conclusions summarize the main findings of this study.

## 2. Data Sources

Wastewater injection volumes (IW) and site location data were obtained from the OCC website http://www.occeweb.com/OG/ogdatafiles2.htm in September 2018 [21] for the calendar years 2006 (reported start date) to 2017. Class II injection and salt water disposal (SWD) volume data sets were manually inspected to remove incomplete or duplicate records, as well as records without geolocation. IW data of SWD wells are available annually from 2006 to 2010, and monthly from 2011 to 2017. Since Osage County, in northeast Oklahoma is regulated by the Environmental Protection Agency (EPA) we could not include all active injection wells to date as these data were not publicly available. However, we obtained information (i.e., location and monthly IW) of 10 active injection wells, within Osage, from Barbour et al [10].

The Oklahoma earthquake database was downloaded from the OGS website http://www.ou.edu/content/ogs/research/earthquakes/catalogs.html [22]. Daily datasets, including epicenter location,

depth and magnitude (i.e., $M_L$, $M_W$, $m_b$ and $M_d$) are available between 1882 to present. In the interest of revealing spatiotemporal patterns of near-recent seismic activity in Oklahoma, only earthquakes occurred after January 2006 are studied in detail.

## 3. Earthquake Unit Homogenization and Data Completeness

### 3.1. Magnitude Unit Homogenization

The type and accuracy of the earthquake recording devices have changed with time. For the most recent decades, despite the instruments remain the same, the used seismic magnitude units vary according to the maximum motion recorded by a seismograph in (1) local magnitude ($M_L$) also known as Richter magnitude, (2) duration magnitude ($M_d$), (3) body-wave magnitude ($m_b$) and (4) moment magnitude ($M_w$). The number of earthquakes occurred between January 2006 and December 2017 is given in Table 1 with respect to the different used magnitude units. Nonetheless, many earthquakes were simultaneously recorded in different scales which facilitate their unit homogenization. As the majority of seismic events (i.e., 25,956) are reported in $M_L$ scale, all other units are converted to $M_L$ to reduce data uncertainty introduced during this conversion. In order to do so, two empirical magnitude conversion relations are derived for those events with significant number of data pairs (i.e., [$M_L$, $m_b$] and [$M_L$, $M_w$]). Since [$M_d$, $M_L$] had zero pairs, a previously derived expression is applied [23] for such a conversion. Scatterplots with the event magnitude pairs, fitted, and 95% confidence envelopes are shown in Figure 1. The derived and used statistical regressions, sample size, cross-correlation coefficients and author are shown in Table 2.

**Table 1.** Number of earthquakes in different magnitude units during 2006–2017 in Oklahoma.

| Magnitude Type | Number of Earthquakes |
|---|---|
| Duration magnitude ($M_d$) | 1763 |
| Body-wave magnitude ($m_b$) | 364 |
| Local Magnitude ($M_L$) | 25,956 |
| Moment Magnitude ($M_w$) | 438 |

**Table 2.** Mathematical regressions adopted and derived to homogenize $M_d$, $m_b$ and $M_w$ seismic magnitudes to local (Richter) magnitude, $M_L$.

| Expression | Sample Size | $R^2$ | Reference |
|---|---|---|---|
| $M_L = 0.936M_d - 0.16$ | 17 | 0.95 | Brumbaugh, 1989 [23] |
| $M_L = 0.85m_b + 0.52$ | 252 | 0.84 | Hong et al (this paper) |
| $M_L = 0.96M_W + 0.35$ | 440 | 0.81 | Hong et al (this paper) |

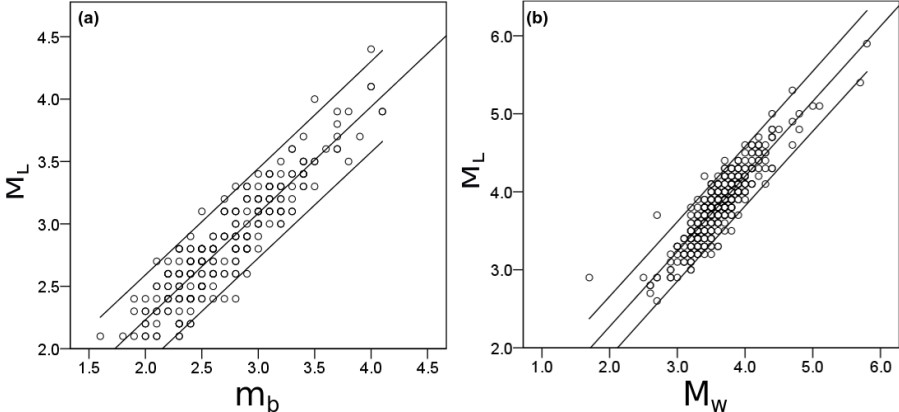

**Figure 1.** Linear regression plots for (**a**) $M_L$ vs. $m_b$ and (**b**) $M_L$ vs. $M_W$ for all seismic events occurred in Oklahoma between January 2006 and December 2017. Conversion equations are shown in Table 2.

### 3.2. Earthquake Magnitude of Completeness

After unit homogenization, the magnitude of all recorded earthquake events in the Oklahoma earthquake catalog, for the period between 2006 and 2017, ranges from 0.1 $M_L$ to 5.9 $M_L$. In an earthquake catalog, the magnitude of completeness ($M_C$) is the minimum magnitude above which earthquakes within a certain region are reliably recorded. Defining $M_C$ is necessary due to the complexity, spatial and temporal heterogeneity of seismometer networks and time series records [24,25]. To assess $M_C$ for our earthquake dataset, a frequency-magnitude distribution (FMD) plot was created for the entire dataset (see Figure 2) based on the entire magnitude range (EMR) method proposed by Woessner and Wiemer [25]. Their method estimates the FMD based on the Gutenberg-Richter law [26]. For the data with magnitude below the assumed Mc, EMR uses a normal cumulative distribution function [25]. Woessner and Wiemer [25] compared the EMR method with other three including maximum curvature-method (MAXC; Wiemer and Wyss, 2000), goodness-of-fit test (GFT; [27]), and Mc by b-value stability (MBS; [28]), and they concluded that EMR is the most favorable model to calculate Mc from regional earthquake catalogues [25]. The FMD curve indicates a data-based suggested value [25] of $M_C$ = 2.6 which will be used as minimum trustable $M_L$ to include in the subsequent analyses.

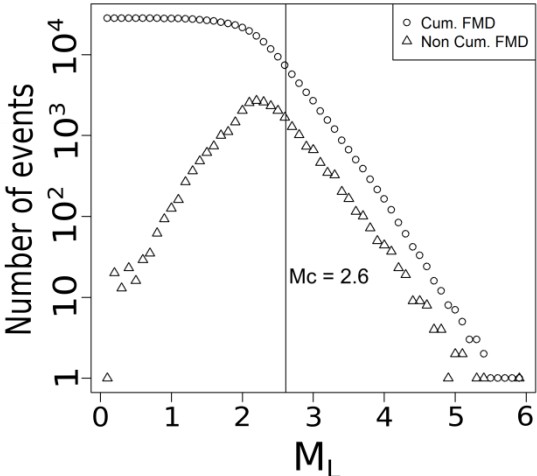

**Figure 2.** Cumulative and noncumulative frequency-magnitude distributions on logarithmic scale with the black line indicating magnitude of completeness ($M_C$) for time series during 2006–2017.

## 4. Interannual Seismicity and Wastewater Injection Activity in Oklahoma

According to the Oklahoma Geological Survey, the earliest recorded earthquake in the state occurred on 22 October 1882 with a $M_L$ 5.0 (Oklahoma Geological Survey 2017). From 1882 to 2002, (120 years) Oklahoma had a total of 186 earthquakes with $M_L \geq M_C$, for an average of 1.55 earthquakes per year (Oklahoma Geological Survey 2017). The Figure 3a,b compile the recent history of earthquake events (bars, N($M_L$)) occurred in Oklahoma from 2000 to 2017 with $M_L \geq M_C$ and discretized by $M_L$ category. Comparatively to its precedent years, the number of seismic events per year with $M_L \geq M_C$ occurred from 2003 to 2008 increased to 4.9 (39 in total; see Figure 3a). However, 2009 appears as a benchmark year that marks a significant increase relative to historic means (see Figure 3b). Between 2009 and 2017, the state had averaged 730 earthquakes per year (6570 total), which is more than four hundred times (i.e., 471) the historic averages up to year 2002. Since "felt earthquakes" usually refer to those with $3 \leq M_W \leq 5$ (National Research Council 2013), Figure 3b also depicts N($M_L$) per $M_L$ category with $M_C \leq M_L < 3$, $3 \leq M_L < 4$ and $4 \leq M_L < 5$ in Oklahoma from 2000 to 2017. A particular peak in seismicity occurred in 2015 with 2560 events, followed by a steady decrease to 786 in 2017. This bell-shaped trend is replicated by each of the used categories in Figure 3b (e.g., $M_C$, 3 and 4),

except by the $M_L \geq 5$ whose peak occurred in 2016. The total number of "damaging earthquakes", which are those with $M_l \geq 5$, also increased after 2009 as shown in Figure 3b.

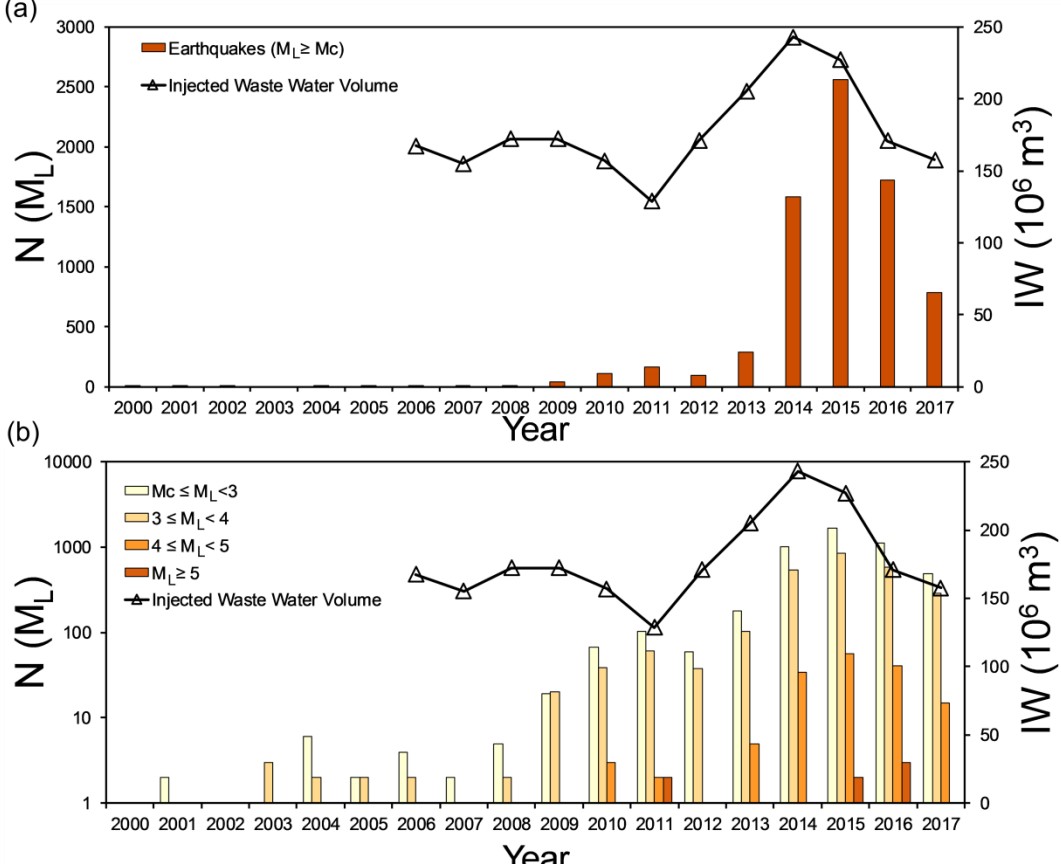

**Figure 3.** (**a**) Time series of total annual number of earthquakes ($N(M_L)$) with $M_L \geq M_C$ (red bars) and oil/gas industry-related injected volumes of wastewater (IW) in million cubic meters (white triangles) in Oklahoma from 2000 to 2017. (**b**) Time series of total annual number of earthquakes ($N(M_L)$) with $M_L \geq M_C$ per magnitude range between 2000 and 2017 and oil/gas industry-related volumes of wastewater injected (IW) between 2006 and 2017 in Oklahoma. Note the log scale for $N(M_L)$ in (**b**).

Crude oil and natural gas have been extracted from Oklahoma's underground for more than 100 years [29]. Between 2010 and 2012 Oklahoma was ranked as the 5th highest producing U.S. state [29]. The high oil and gas production rates caused a rapid increase in construction of underground injection Class II (UIC) wells, widely used to enhance the recovery of oil (EOR Enhanced Oil Recovery wells) and disposing of industrial wastewater (SWD) since the 1930s [30]. Figure 3a,b both show IW (in $10^6$ m$^3$/year) from OCC UIC Class II wells reports that begin in 2006 to near present. From 2006 to 2012 the volume of injected water ranged around $150 \times 10^6$ m$^3$/year. However, starting in 2012 a rapid increase in IW volumes is observed that peak in 2014 and 2015, followed by a sharp decline in 2016 and 2017 when the IW gets back to a number around the 2006–2012 average of $150 \times 10^6$ m$^3$/year. A paired time series analysis of the coupled IW and $N(M_L)$ reveals that both variables have shown a similar trend since the start of the unconventional use of injected water to retrieve oil and gas.

Regionally, the spatial distributions of earthquakes occurred in Oklahoma between 2006 and 2017 with $M_L \geq M_C$ and the corresponding location of wastewater disposal wells operated during the most active year (i.e., 2014) are illustrated in Figure 4a,b. In both figure panels, the different symbol sizes represent different categories of $M_L$ and IW. The spatial distribution of the two variables resembles a spatially correlated structure whose dependency functions need to be determined for different time lags. Further, during this period (2006–2017) most earthquakes occurred in central and northern

Oklahoma while the largest magnitude ones occurred in the central region of the state. Accordingly, historically the largest IW volumes occurred mainly in central and northern Oklahoma. In counties like Osage, seismicity appears to be low possibly due to the dense rock bodies that reduce seismogenic potential for basement faults [31,32].

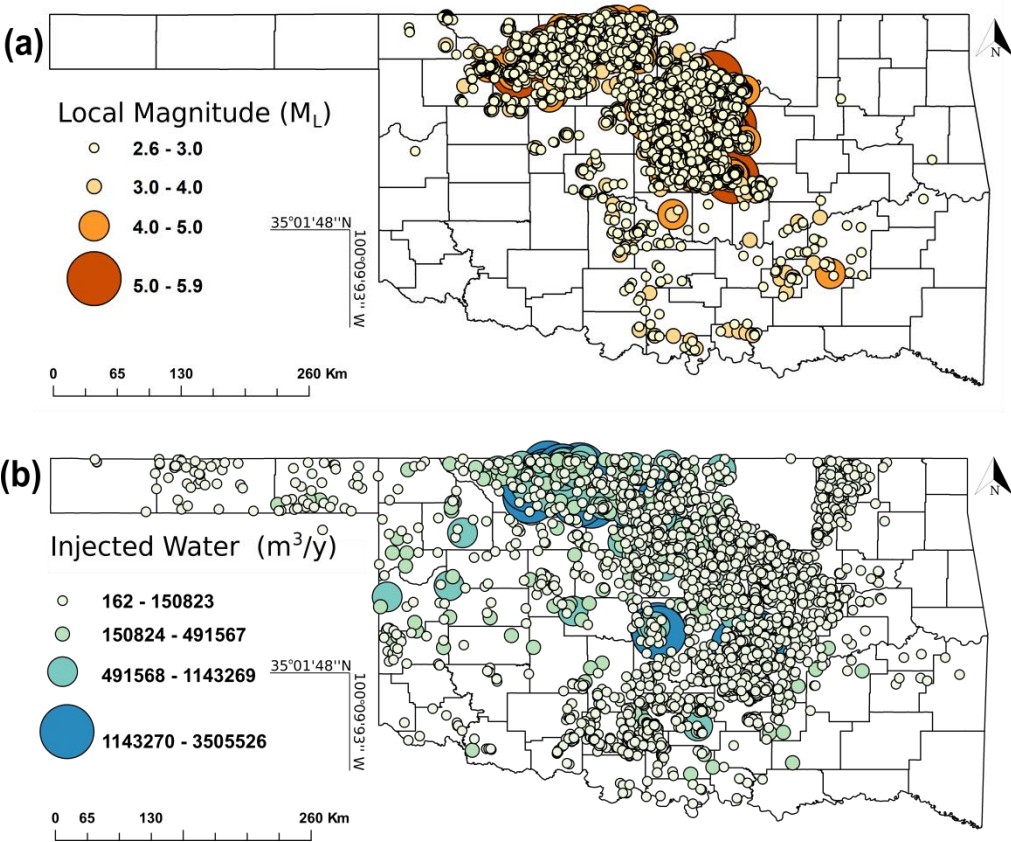

**Figure 4.** (**a**) Spatial distribution of earthquakes with $M_L \geq M_C$ occurred in Oklahoma from 2006 to 2017; (**b**) Spatial distribution of wastewater disposal wells with corresponding IW volume (m$^3$/year) operated in 2014.

## 5. Regional Migration Pattern of Epicenters and Wastewater Injection Activity

Since the spatial distribution of earthquakes appears to be highly conditioned by the zonal intensity of underground water injection, as shown in Figure 4, a cluster analysis can provide a clearer picture of the spatial co-variance between the two processes in play. Figure 5a shows the spatial distribution of weighted mean centers and standard deviational ellipses of all recorded Oklahoma earthquake epicenters occurred during each year from 2006 through 2017. A weighted mean center $(X_T, Y_T)$ in any year (T) is the representative geographic location of all epicenters $(X_i, Y_i)$ adjusted for the local magnitude $M_L$ associated with each earthquake (i) acting as weighting factors $(w_i)$ as shown in Equation (1) [33].

$$\overline{X_T} = \frac{\sum_{i=1}^{n} w_i X_i}{\sum_{i=1}^{n} w_i} \qquad \overline{Y_T} = \frac{\sum_{i=1}^{n} w_i Y_i}{\sum_{i=1}^{n} w_i} \tag{1}$$

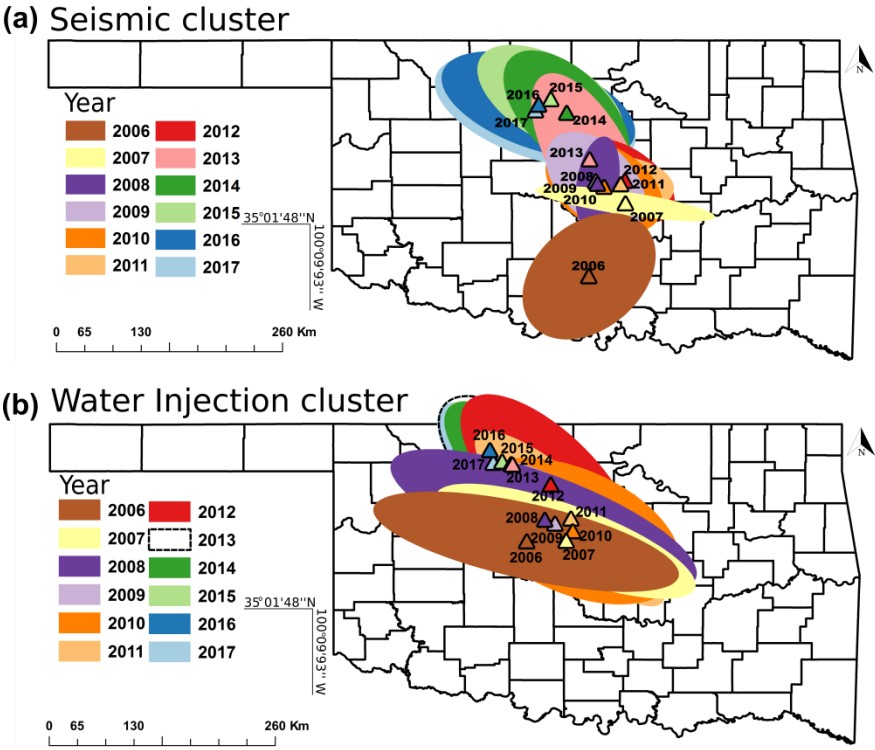

**Figure 5.** (**a**) Earthquake-clustering occurrence by year. Epicenters' weighted mean centers (triangles) and standard deviation ellipses of all recorded earthquakes occurred in Oklahoma between 2006 and 2017; (**b**) Wastewater injection volume weighted mean centers (triangles) and standard deviation ellipses in Oklahoma between 2006 and 2017. The colors in both panels match for the same years, except by 2013 whose dashed lines are intended to improve result visualization. Coordinates of mean weighted centers are computed using Equation (1).

Where $w_i$ is the $M_L$ for each earthquake event (i) in a particular year T. Following equation (1), weighted mean centers of all earthquakes occurred in a particular year T would be closer to epicenters with the largest $M_L$ during that year. The major and minor axes of these weighted standard deviation ellipses are calculated as the second moment of the x- and y-coordinates distribution from each weighted mean center [34]. This cleaner approach, illustrated by Figure 5a, shows a generalized northwest seismic migration pattern from 2006 through 2017. Correspondingly, Figure 5b illustrates the weighted mean centers and standard deviation ellipses of wastewater disposal wells in each year from 2006 through 2017. Analogously to epicenters, wastewater injection locations are weighted by the volumetric magnitude of the annual injection volumes associated with each well. Thus, the weighted mean center of wastewater disposal wells in a particular year would be geographically closer to wells with larger annual injection volumes, reflecting the regional trend of well activity in that specific year. In summary, both unconventional oil and gas extraction and earthquake count show a northwest migration pattern during 2006 to 2017. To recognize year to year migration patterns, Figure 6 compiles those bivariate trends through vectors whose length is proportional to the average migration distance between consecutive years. The diagram shows some years when both processes migrated similar distances in similar directions, particularly 2007–2008 (~33 to 35 km SE), 2009–2010 (10 to 20 km NNE), 2012–2013 (~43 km SSE), 2014–2015 (~9 to 22 km SE) and 2016–2017 (7 to 15 km W). In other cases, the two vectors keep an angular distance greater than 90 degrees such as 2006–2007, 2010–2011 (N-E quadrant), 2011–2012 (mostly E quadrants), 2015–2016 (mostly S quadrants). The large disparity in distances in 2006–2007, 2011–2012 and 2013, 2014 maybe due to the fact that injection operations moved quickly in the last months of the last year and earthquake count (since it has a lagged response) did not immediately showed the expected pattern of migration. Overall, regional migration patterns seem

to correspond to one another evidencing a zonal effect of the unconventional oil and gas industry on the number of regional earthquake count.

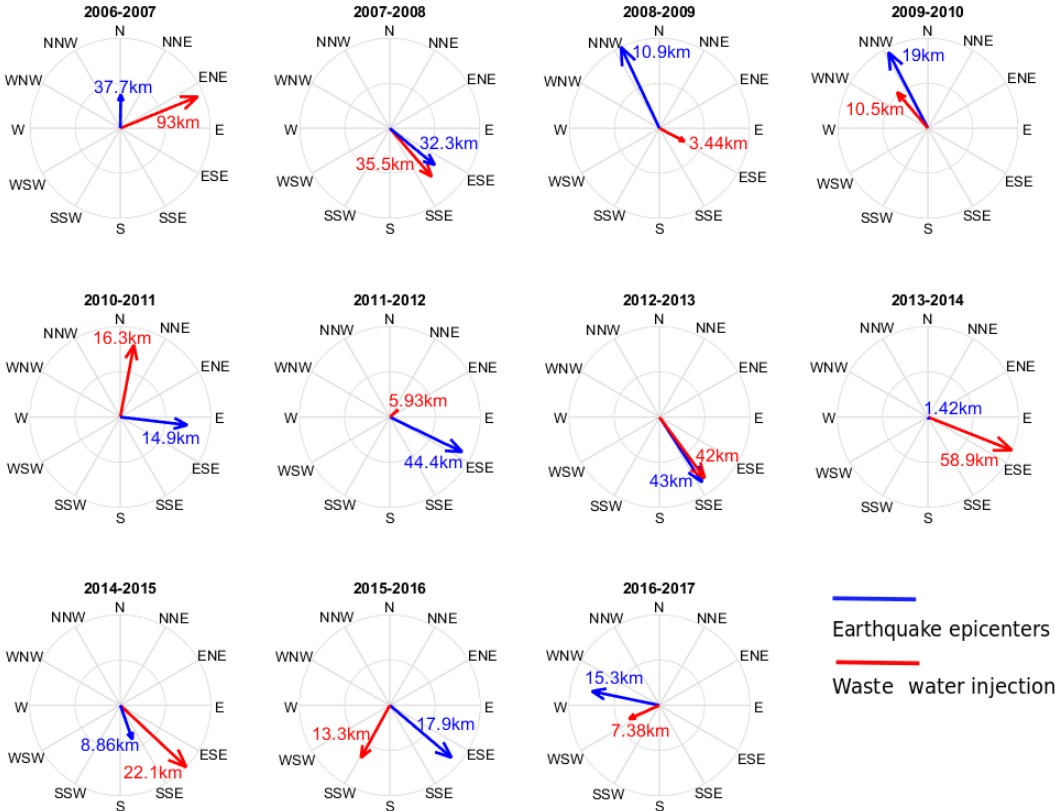

**Figure 6.** Yearly migration patterns between earthquakes weighted epicenters and wastewater injection activity in Oklahoma since 2006. Red and blue lines mean the average displacement of mean weighted centers of wastewater injection and earthquakes between consecutive years. The average displacement distance is also indicated within each compass diagram.

## 6. A Parsimonious Model of Seismicity.

With the objective of proposing a (single-variable) parsimonious regional predictive model between cumulative wastewater injection (IW) and earthquake count (N), patterns of lagged seismic responses to cumulative injected water during the 2006 to 2017 period in Oklahoma are explored. A cross-correlation analysis is carried to determine the temporal lags (i) seemingly to mostly control the number of expected earthquakes in a particular month t ($N_t$) as a function of $IW_{t-i}$ for i = 0, 1, 2, etc, months. This time delay (i.e., i) can be physically expressed as the time the pressure increase takes to propagate from the injection wells to critically stressed faults in the crystalline basement [15,18]. The cross-correlogram illustrated in Figure 7a reveals that lags i = 0 through −25 previous to the seismic events appear to mostly contribute to the bivariate co-dependence between IW (predictor) and N (predictand). The Figure 7b quantifies the contribution of each lag i to the total correlation structure above the Pearson correlation coefficient significance threshold. According to the correlations for lags 0 to 25 months, we extract weight coefficients ($w_i$) for each lagged contribution to express $\hat{IW}$ as a function of $IW_{t-i}$ (i = 0, 1, 2 . . . , 25 months) as shown in Equation (2):

$$\hat{IW} = \sum_{0}^{25} w_i IW_{t-i} = w_0 IW_t + w_1 IW_{t-1} + w_2 IW_{t-2} + \ldots w_{25} IW_{t-25} \tag{2}$$

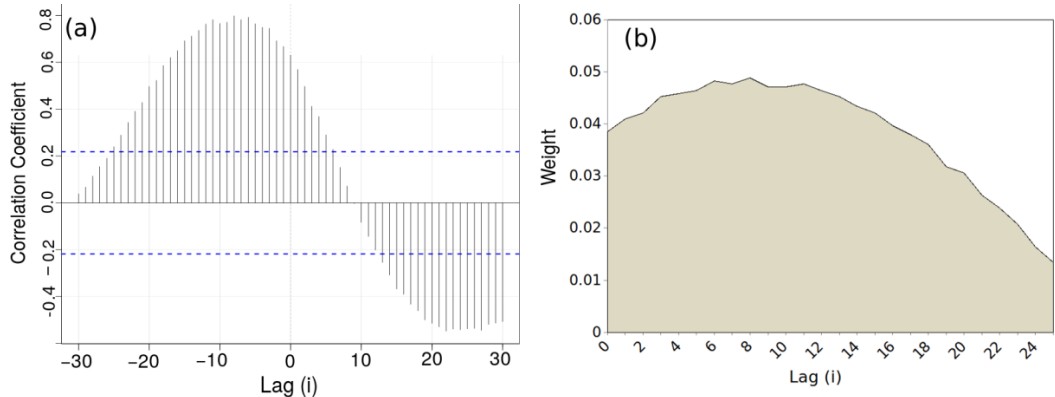

**Figure 7.** (**a**) Cross-correlation diagram between $IW_{t-i}$ and $N_t$ for different lags of IW (e.g., i = 0, 1, 2, 3 . . . , n months). Negative numbers mean that IW precedes $N_t$. (**b**) Contribution ($w_i$) of each lag i to the prediction of the total of number of earthquakes in a particular month t ($N_t$), to be applied to the predictors in Equation (2).

From Figure 7b it can be expected that lags 0 to 10 will be responsible for 50% of the variability in $I\hat{W}$ in Equation (2), confirming that the number of earthquakes in a particular month is the result of the pressure buildup due to previous-months water injection activity. According to the correlation structure derived, we fitted a mathematical power-law relating $I\hat{W}$ ($m^3$/month) to $N_t$ for all $M_L \geq M_C$ (see Equation 3). Figure 8 illustrates such a fitted relation applied to the logarithms of the monthly values since 2011 (complete data pairs).

$$N_t = 3.2099 \times 10^{-6}\ I\hat{W}^{6.1489} \tag{3}$$

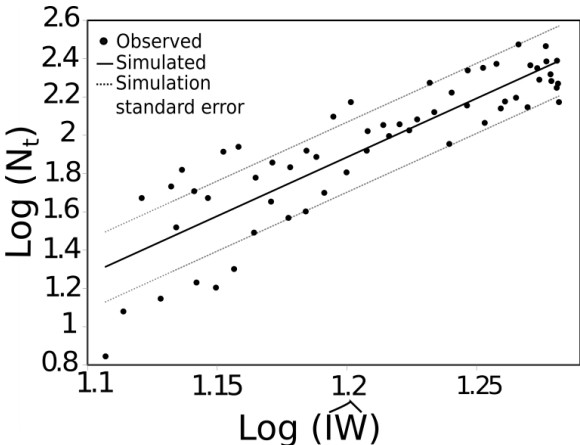

**Figure 8.** Regional induced-earthquake count $N_t$ ($M_L \geq M_C$) and $I\hat{W}$ estimator calibrated between years 2006 and 2017 in the state of Oklahoma. The power law explains 77% of the bivariate behavior of monthly injection and earthquakes number. Upper and lower dashed lines representing standard errors of estimates have been added to the mean predicted values.

We found that a power law of this type retrieves the highest coefficient of determination and explains 77% of the bivariate dependency between weighted cumulative regionally injected water and seismicity in north-central Oklahoma. Standard error lines also provide a statistical estimation of the average error when using this relationship in a predictive manner. Figure 8 also helps extracting some interesting numbers to compare with historical benchmark periods with distinct seismic (e.g., number of events) activity in Oklahoma (see Section 4; Oklahoma Geological Survey, 2017). By applying this relationship to a scenario of hypothetical, constant-in-time, injection rate, we can compare with historic

benchmarks and study possibilities for sustainable oil and gas extraction limits (see Table 3) in terms of the expected number of seismic events (i.e, $N_t$). The term "sustainable limit" in column 5 of Table 3 refers to potential maximum injection values per month that the Oklahoma state regulation authorities (e.g., OCC and/or EPA) could consider for regulation of the oil and gas industry. This by no means accounts for other influences on environmental issues like water or energy consumption, groundwater, land or air pollution. In this table, it appears that constant and continuous rates (i.e., $IW_{t-i}$) of 5.6 million m$^3$/month (i = 0, 1, 2, 3, etc) could reduce the number of earthquakes to pre- year 2000 conditions to 1.55 earthquakes per year with magnitude $M_L > M_C$. However, one could also define other limits like the mean water injection during the period 2003–2008 (6.8 million m$^3$/month) as a sustainable limit, but at the expense of potential additional seismic occurrences (5.1 events/year) similar to the beginning of the 2000 decade or previous to the boom of oil and gas extraction (pre 2009–2015). Also note how, since this law is potential, an increase in $1 \times 10^6$ m$^3$/month of injected water represents different changes in seismicity across the spectrum of IW values with larger values triggering dramatic increases in seismic events, $N_t$. As an example, an increase of 1 million m$^3$/month above 19 million m$^3$/month (super-boom scenario) would represent more than 1000 additional earthquakes ($M_L > M_C$) per year.

**Table 3.** Predicting $N_t$ (number of earthquakes/year) in terms of hypothetical scenarios of different weighted average ($I\hat{W}$; Equation (2)) or monthly constant IW in light of historical records and benchmark periods. Uncertainty interval estimates have been added to each predicted $N_t$. Historical benchmark periods have been extracted from section 4 this manuscript for reasons of comparison.

| $I\hat{W}$ ($\times 10^6$ m$^3$/month) | $N_t$ (number/year) | $N_t$ Interval [min, max] (number/year) | Historical Benchmark Period | Sustainable Limit? |
|---|---|---|---|---|
| 1 | $3.5 \times 10^{-5}$ | $2.53 \times 10^{-5}$, $5.87 \times 10^{-5}$ | - | - |
| 3 | 0.03 | 0.02, 0.05 | - | - |
| 5 | 0.76 | 0.50, 1.17 | - | - |
| 5.6 | 1.54 | 1.01, 2.34 | 1884–2002 | Pre- 2002 |
| 6.8 | 5.07 | 3.32, 7.21 | 2003–2008 | Pre oil and gas boom (2003–2008) |
| 7 | 6.05 | 3.97, 9.23 | - | - |
| 9 | 28.4 | 18.6, 43.3 | - | - |
| 11 | 97.5 | 64.0, 148.6 | - | - |
| 13 | 272 | 179, 415 | - | - |
| 15 | 657 | 431, 1001 | - | - |
| 15.2 | 712 | 467, 1086 | 2009–2017 | Peak period |
| 15.4 | 788 | 517, 1200 | 2017 | Oil/gas price fall/OCC regulation |
| 17 | 1417 | 930, 2161 | - | - |
| 18.7 | 2547 | 1671, 3882 | 2015 | Peak year |
| 19 | 2809 | 1843, 4281 | - | - |
| 20 | 3851 | 2527, 5869 | - | - |
| 21 | 5198 | 3411, 7922 | - | - |
| 23 | 9095 | 5968, 13861 | - | - |

## 7. Model Output Intercomparison

This section offers a model result intercomparison of our parsimonious approach with two of the recently developed predictive models by Norbek and Rubinstein [17] and Langenbruch and Zoback [18]. All models use monthly injection rates as predictand and a number of time series-derived or geology-inferred parameters. For this intercomparison, we used the database published in Norbek and Rubinstein [17] on monthly injection rates, observed seismicity (declustered catalog) and the model outputs from the Hydromechanical [17] and Seismogenic [18] models. Since Norbek and Rubinstein [17] used different magnitude of completeness ($M_C = 3.0$) and removed (declustered) any instances of foreshocks and aftershocks from the main events, the parsimonious model had to be re-calibrated for this data-based but maintaining the $IW_{t-i}$ correlation derived in Section 6 of this

manuscript. For an $M_C = 3$ and declustered database, Equation (4) represents the number of expected main shocks as a function of the antecedent 25 months of waste water injection (using Equation 2). Similarly to Equation (3), this model explains 75% of the seismic activity at a monthly time scale.

$$N_t = 7.9630 \times 10^{-3} \hat{IW}^{3.4556} \tag{4}$$

Figure 9 illustrates the results of such a model intercomparison in light of monthly injection rates between 2008 and 2018, including observed and predicted seismicity values from the three models. As noted by Norbeck and Rubisntein [17] although the hydromechanical model outputs seem to capture the general long-term trends, after year 2016 the model overestimates seismic activity, showing some weakness in capturing sharp changes in water injection. The seismogenic model seems to capture such short term variability more thoroughly but tends to under-predict in times of low water injection. The parsimonious model proposed in this article seems to capture both low and high seismic activity but fails at capturing short term sharp variability in a similar fashion to the hydromechanical model.

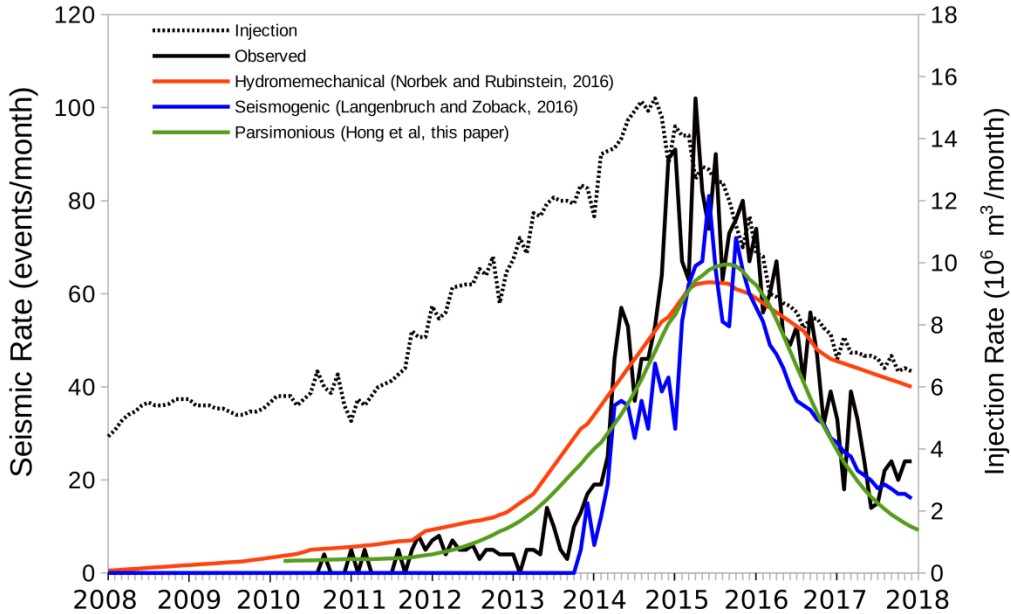

**Figure 9.** Model intercomparison experiment using the hydromechanical, seismogenic and parsimonious models for retrospective simulations of seismicity in Oklahoma between 2008 and 2018 in light of observed (declustered) seismic events and monthly waste water injection rates.

## 8. Discussion

### 8.1. Acknowledging Methological Limitations

The results from this study need to be understood in light of the data and methodology limitations of our analyses. First, results mainly focus on statistical spatiotemporal relationships between wastewater injection volumes and earthquakes number and magnitude. Second, since the magnitude unit conversion (e.g., $M_w$ to $M_L$) procedure introduces a maximum uncertainty of 19% (see Section 3.1 and Table 2) the location and size of the weighted mean centers and standard deviation ellipses will have a maximum inherited error of 9.5%. Third, the analyses did not consider other influences on earthquakes' induction or generation mechanisms such as regional rock fracturing or geologic structures that propagate or moderate seismic waves. Moreover, due to the limiting number of years with data (2006–2017), we do not know how the panoramic would look like in the future in views of higher (or lower) levels of wastewater injection. Further we recommend caution when planning to use of the statistical relationships found here for future years as the rock systems might not behave in a linear fashion anymore since the increasing rock-fracturing processes might propagate

across larger regions becoming a network of interconnected faulted systems that might translate in widespread earthquakes swarms. Finally, the results achieved in this study, however, need to be further explored within different subregions to consider particular geological heterogeneities that could result in potentially different behaviors than the ones shown here.

*8.2. Contributions to State-of-the-Art*

The conducted spatiotemporal analyses and proposed parsimonious model represent a novel contribution for prediction, model intercomparison and decision making. In terms of process understanding, the results from this manuscript are clear to relate the geographic scope and lagged dependency between wastewater injection volumes and earthquake count. Second, if used as stated, they can help predict the number of earthquakes in a particular month in terms of the antecedent monthly injection volumes. What we can define as sustainable extraction limit (conditions pre year 2000) could be 5.6 million $m^3$/month or any combination of values of IW during the antecedent 25 months that allow obtaining around 1.5 earthquakes per year with $M_L \geq M_C$. A similar number of $5 \times 10^6$ $m^3$/month was also found by Langenbruch and Zoback [18]. However, these authors propose a steady injection condition per month, but according to Equation (2) there could be other combinations of differential (seasonal) injection that could lead to the same result of minimum earthquakes. Third, possibly the best utility of the results of this manuscript is its use as a tool for model intercomparison with current and future models. For example, Pollyea et al. [20] developed a geospatial analysis of the bivariate occurrence of earthquakes with the location of salt-water disposal wells. We coincide with Pollyea et al. [20] in that there is a general of north-west migration of both processes. However, a difference we find is the fact that instead of circles we obtained the two-axis variability ellipses and weighted the well and epicenter locations by magnitude and injection volumes, providing a more accurate description of their spatially correlated distribution. We also provide year by year direction of migration and distance patterns. Results from the model output intercomparison experiment show comparable capabilities of the Parsimonious (Hong et al, this paper) and Hydromechanical [17] models in the long-term with the Parsimonious representing better the recent decline in seismicity conditions. However, both these models seem to have a weak performance at detecting rapid changes better captured by the seismogenic model [18].

*8.3. Contributions to Sustainable Extraction and Decision Making: What are Sustainable Limits?*

Due to the earthquake upsurge since 2009, the Oklahoma Corporation Commission adopted a "traffic light" system since 2013 in response to the concerns over underground fluid injection induced earthquakes. In a "traffic light" system, if no underground fluid injection induced earthquakes occur, operators could continue their injection activities at regulated rates under a green light condition. Once an earthquake occurs, operators are under yellow light condition. They should investigate the relationship between the earthquake and injection activities and reduce injection rates. If an earthquake event induced by underground injection occurs and the triggered seismicity cannot be stopped by reducing injection rates, operators are under the red light condition and should be prepared to terminate injection activities [20,35,36]. The OCC "yellow light" permitting system requires operators to monitor for background seismicity and shut down wells to record bottom hole pressure every 60 days. The Oklahoma Corporation Commission has been evolving the "traffic light" system applications based on updated research results and new data [12]. The slight decrease in earthquake occurrence in 2016 and 2017 (Figure 3) has been attributed to these mitigation efforts. However, as noted by Pollyea et al. [20] these decreases could have also been the result of the dramatic drop in oil prices. The results presented in this manuscript could be used as a parsimonious cause-effect method and whose results could be used to potentially improve the current "traffic light" policy, inform legislators and decision makers by providing sustainable limits for oil and gas extraction in order to minimize the expected number and magnitude of subsequent quakes, thus avoiding future human and property losses. The availability of more information for upcoming years will serve to provide robustness, not only to this, but to other

current methods with the main purpose to raise conscience of the potential of human-induced seismic activity and balance out gains for economy, environment and society.

## 9. Conclusions

This study has gathered comprehensive datasets of oil and gas industry-related wastewater injection volumes and earthquakes number with associated event magnitudes from 2006 to 2017 over the entire state of Oklahoma. Data were analyzed to remove those seismic events below the threshold of magnitude completeness. First, we explore the spatiotemporal variability of both processes and conclude about a high correspondence between the two that further supports the hypothesis that the recent boom in oil and gas production through unconventional methods with wastewater injection was potentially responsible for the upsurge in the state's seismic activity during 2006 through 2015. Also, a reduction in the number of earthquakes per year, in years 2016 and 2017, reflect either the mitigation policies dictated by OCC or the drop in oil and gas prices or both. Second, a cluster analysis reveals a correlated migration pattern between earthquake occurrences and salt water injection activity. Following the migration of the weighted wastewater injection ellipses, weighed epicenters show a predominant northwest direction pattern during the 2007–2017 period. Third, a lagged cross-correlation analysis shows that first, the number of induced earthquakes in a subsequent month is strongly associated with the previous 25-month cumulative wastewater injection volume and a power law can be fitted between number of quakes and weighted average monthly injection volumes as predictive tool with a coefficient of determination of $R^2 = 0.77$. Using such a relation, several sustainable extraction limits are explored and compared with historic means. Results from these analyses coincide and expand on previously sustainable limits of 5 to 6 million $m^3$/month to potential combinations that could attain the same number within the 25 previous months. A model intercomparison of our parsimonious model, a hydromechanical model, and a seismogenic model reveals a satisfactory performance of the proposed approach and similitude to the hydromechanical model outputs. Nonetheless monthly sharp changes in seismicity could only be more appropriately represented by the seismogenic model. The approach proposed in this manuscript could potentially be regionalized according to the geology of each zone and results could potentially be used as a tool for further model intercomparison experiments and decision making on spatially varied permission distribution and regional industry development to minimize negative consequences of induced earthquakes.

**Author Contributions:** Conceptualization, Z.H., H.A.M. and Y.H.; Methodology, Z.H.; Software, Z.H.; Formal Analysis, Z.H. and H.A.M.; Writing and Original Preparation, Z.H.; Writing—Review and Editing, H.A.M.; Visualization, Z.H.

**Funding:** This research received no external funding.

**Acknowledgments:** The earthquake catalog was obtained from the Oklahoma Geological Survey (http://www.ou.edu/content/ogs/research/earthquakes/catalogs.html), UIC (Underground Control) wells data were obtained through the Oklahoma Corporation Commission (http://www.occeweb.com/og/ogdatafiles2.htm). The authors thank Laura Labriola for her valuable proofreading.

**Conflicts of Interest:** The authors declare no conflict of interest.

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
