# Peer review of "Spatiotemporal Assessment of Induced Seismicity in Oklahoma: Foreseeable Fewer Earthquakes for Sustainable Oil and Gas Extraction?"

_geosciences, doi:10.3390/geosciences8120436_

Round 1
Reviewer 1 Report
After looking over the revised manuscript and the authors’ covering letter, I am satisfied that the authors have responded to my somewhat harsh review comments quite comprehensively. I have a few additional suggestions, however.
1. The authors claim that their statistical results apply to Oklahoma. This is not entirely true. Figures 4 a and b show a big blank area with neither seismicity nor wastewater disposal. The blank area is Osage County, which has lots of wastewater disposal and some seismicity. I’m sure the authors must know that this area is blank because the EPA, not the OCC, has regulatory authority there. This oversight is of some consequence, especially as injection wells in Osage probably contributed to the 2016 M5.8 Pawnee earthquake.
2. Line 237 – I could not find the reference for Simpson et al. (1976).
3. I feel obliged to note that it is sorely in need of careful proof reading. It has a great many grammar and spelling errors.
Author Response
Please attached PDF and refer to the section Reviewer #1

Reviewer 2 Report
This paper is much improved from the original submission, particularly with the addition of the catalog completeness threshold and the additional discussion and comparison with other models. I appreciate the thoroughness with which the catalog magnitudes are homogenized and the completeness magnitude is now computed. However, I believe there are still a few shortcomings that could be addressed to make it a stronger paper.
1) The addition of the completeness magnitude definitely helps make the results more believable. One further question though – did you examine changes in the completeness magnitude over time? Because of changes in the number and geometry of the stations over time, I would be surprised if the catalog was complete all the way down to M2.6 from 2006-2009. A more typical (and conservative) choice for a constant completeness magnitude that is valid from 2006-2017 would be M3.
2) I think Figure 8 and Equation 3 could use a bit more detail. First, am I correct in assuming that N_t is actually the log of the number of events? And is IW really the log of the injected water volume? Please clarify that, the way it is written now is a bit confusing. Second, what are the uncertainties in this regression? Can you put error bars in Figure 8 and/or the numbers of events reported in Table 3? I am also having a bit of difficulty understanding Table 3. What do you mean by “Benchmark period”? How did you determine these periods, and why is 1884-2002 separate from 2003-2008 (also, again, I would be very surprised if the catalog was complete down to M2.6 in the 1884-2002 time period). And what exactly does the “Sustainable limit?” column mean? I think you could explain this table in more detail in the text.
3) Section 7 has certainly benefitted from the addition of subsection 7.2 and discussion of alternative models. At the end of the section (ll. 321-324) you make some claims about what your model offers over the other models. Have you thought about quantitatively testing this, either pseudo-prospectively or retrospectively? For example, comparing your forecasts with the other model forecasts, along the lines of what Norbeck and Rubinstein (2018) do when comparing their model with the USGS model and the Langenbruch and Zoback (2016) model.
4) In general, you need to be a bit careful with the use of the term “swarm”, which has a more specific meaning than you seem to utilize in this manuscript. A swarm is a specific type of earthquake sequence, and while some of the earthquake sequences that have occurred in Oklahoma since 2009 are likely earthquake swarms (e.g., the 2009 Jones swarm), much of the seismicity consists of standard mainshock-aftershock (or foreshock-mainshock-aftershock) sequences with perhaps higher aftershock triggering than they would have had prior to the increase in injection (e.g., Llenos and Michael, 2013). In particular, the use of the term in ll. 271-272 seems wrong, it suggests that Nt is a number of sequences (swarms), rather individual earthquakes, which is how I think you actually mean it. Similarly, ll. 358-359.
Minor comments
ll. 63-66: I would argue that such assessments do in fact exist (e.g., Langenbruch and Zoback, 2016; Pollyea et al., 2018), which explicitly discuss the effects of decreasing injection on the spatial and/or temporal patterns of earthquakes. You may also find this recently published paper of interest:
Langenbruch, C., M. Weingarten, and M. D. Zoback (2018). Physics-based forecasting of man-made earthquake hazards in Oklahoma and Kansas, Nature Comm., 9, doi: 10.1038/s41467-018-06167-4.
ll. 220-222: This sentence is written in a somewhat confusing matter. Why would a physically-based study need to prove that your data did NOT necessarily reflect that hypothesis? Also, are there studies that you can cite here that suggest that the earthquakes may be spreading beyond the zonal areas of activity? This sounds like a very specific hypothesis.
l. 237: The Langenbruch et al. (2017) and Simpson et al. (1976) papers cited here seem to be missing from the reference list. Are the time delays observed here physically reasonable?
Author Response
Please see the attached PDF document and refer to Reviewer #2 section.

Reviewer 3 Report
Comments can be found from the annotated pdf file of the manuscript.

Author Response
Please see the attached PDF document and refer to section titled Reviewer #3

Round 2
Reviewer 3 Report
Please change Ms to Md in Table 1